# The Effect of Complex Alleles of the *CFTR* Gene on the Clinical Manifestations of Cystic Fibrosis and the Effectiveness of Targeted Therapy

**DOI:** 10.3390/ijms25010114

**Published:** 2023-12-21

**Authors:** Maria Krasnova, Anna Efremova, Artem Bukhonin, Elena Zhekaite, Tatiana Bukharova, Yuliya Melyanovskaya, Dmitry Goldshtein, Elena Kondratyeva

**Affiliations:** Research Centre for Medical Genetics, Moscow 115522, Russia; krasnova.m.g.0605@gmail.com (M.K.); a.v.bukhonin@gmail.com (A.B.); elena_zhekayte@mail.ru (E.Z.); bukharova-rmt@yandex.ru (T.B.); melcat@mail.ru (Y.M.); dvgoldrm7@gmail.com (D.G.); elenafpk@mail.ru (E.K.)

**Keywords:** complex alleles, cystic fibrosis, *CFTR* gene

## Abstract

The authors of this article analyzed the available literature with the results of studying the prevalence of complex alleles of the *CFTR* gene among patients with cystic fibrosis, and their pathogenicity and influence on targeted therapy with CFTR modulators. Cystic fibrosis (CF) is a multisystemic autosomal recessive disease caused by a defect in the expression of the CFTR protein, and more than 2000 genetic variants are known. Clinically significant variants are divided into seven classes. Information about the frequency of complex alleles appears in a number of registers, along with the traditional presentation of data on genetic variants. Complex alleles (those with the presence of more than two nucleotide variants on one allele) can complicate the diagnosis of the disease, and change the clinical manifestations of cystic fibrosis and the response to treatment, since each variant in the complex allele can contribute to the functional activity of the CFTR protein, changing it both in terms of increasing and decreasing function. The role of complex alleles is often underestimated, and their frequency has not been studied. At the moment, characteristic frequently encountered complex alleles have been found for several populations of patients with cystic fibrosis, but the prevalence and pathogenicity of newly detected complex alleles require additional research. In this review, more than 35 complex alleles of the *CFTR* gene from existing research studies were analyzed, and an analysis of their influence on the manifestations of the disease and the effectiveness of CFTR modulators was also described.

## 1. Introduction

Cystic fibrosis (CF) is an autosomal recessive disease that affects approximately 1 in 6000 newborns among the majority of the Caucasian race [1], although its frequency may vary in certain groups. CF is caused by the presence of pathogenic variants of the cystic fibrosis transmembrane conductance regulator (*CFTR*), which encodes a transmembrane protein consisting of 1480 amino acids and forms a cAMP-dependent chloride channel in the membranes of epithelial cells [2].

First of all, the respiratory system, pancreas, intestines, and sweat glands are affected with the disease. Impaired mucociliary clearance due to the formation of thick mucus causes the obstruction of the respiratory tract in combination with chronic bacterial infection, which results in the development of respiratory failure, which is the main cause of mortality. In more than 85% of patients, the pancreas does not produce digestive enzymes, which leads to chronic pancreatic insufficiency [1,3].

Since the identification of the *CFTR* gene in 1989, more than 2000 variants of its nucleotide sequence have been described [2]. A significant variation in the phenotypic manifestations of CF in patients may be due to the action of a large number of factors, including the diversity of *CFTR* genotypes, the influence of modifier genes, and environmental factors. As of 7 April 2023, on the website of the CFTR2 international project (https://cftr2.org, accessed on 14 July 2023) 719 pathogenic genetic variants of *CFTR* are presented. These variants interfere with the synthesis of the CFTR protein and its transport to the apical membrane of the cell, or disrupt its function as a chloride anion channel. The Russian register of patients with CF for 2021 contains information on 233 pathogenic variants; 132 of them occur repeatedly, and 83 genetic variants are absent from international CFTR databases [4]. The three most common variants of the *CFTR* gene nucleotide sequence in Russia are p.Phe508del (F508del) (frequency 51.55%), CFTR2,3dele (21 kb) (frequency 6.11%), and p.Glu92Lys (E92K) (frequency 3.46%). In 1993, Welsh and Smith classified variants depending on their predicted or experimentally evaluated functional effect on the synthesis and function of the CFTR protein for the first time [5,6]. Pathogenic variants causing the disruption (absence) of *CFTR* mRNA formation (Class 7), and disrupting the synthesis (Class 1) and folding of CFTR protein (Class 2), lead to the formation of critically small amounts of CFTR protein. Class 3 variants result in the synthesis of normal amounts of CFTR protein at the cell membrane, but the defect gating regulation of CFTR disturbs the channel function. Class 1–3 variants lead to a complete loss of protein synthesis; in other words, these classes of mutations are associated with classical manifestations of CF [3]. Variants that cause a disruption of the conductivity of the CFTR channel (Class 4), the formation of a reduced amount of CFTR protein (Class 5), or the formation of an unstable protein that is prematurely recycled from the apical membrane and degrades in lysosomes (Class 6) will lead to a partial decrease in the functional activity of the CFTR channel, which is often associated with less severe CF phenotypes [7].

CF has shifted from a “fatal” disease to the chronic disease category, but the median life expectancy in developed countries ranges from 20 to 40 years or more [8,9]. The most significant result of the development of CF pharmacotherapy is the modern strategy of targeted therapy with CFTR modulators aimed at restoring the structure and function of the CFTR protein. For the treatment of CF, four targeted drugs have been registered by FDA and EMA, consisting of a potentiator ivacaftor and correctors, namely, lumacaftor, tezacaftor, and elexacaftor, in different combinations that eliminate specific defects of the CFTR protein [10]. In the case of the most common genotype, p.Phe508del/p.Phe508del, a functional recovery of CFTR is observed with the use of all three combined drugs, i.e., lumacaftor/ivacaftor, tezacaftor/ivacaftor, and elexacaftor/tezacaftor/ivacaftor) [10,11].

Today, the growing number of newborn screening programs and the use of sequencing often lead to the detection of rare variants of the *CFTR* gene or to the identification of complex alleles of the *CFTR* gene [12]. Complex alleles arise as a result of a combination of two or more *CFTR* variants in the cis position on the same allele. The combination of several variants can affect individual stages of CFTR protein biogenesis; therefore, complex alleles can have a strongly pronounced pathogenic effect, whereas each individual variant included in the complex allele may have a “mild” effect or none at all. For example, such an additive effect is shown for p.[Arg74Trp;Val201Met;Asp1270Asn] ([R74W;V201M;D1270N]), while the variant p.Arg74Trp is a polymorphism, and the variants p.Asp1270Asn and p.Val201Met cause a decrease in protein function [13]. One variant of p.Asp1270Asn (in combination with another pathogenic parent allele in the compound-heterozygote state) is not enough to cause CFTR channel dysfunction. The p.[Arg74Trp;Asp1270Asn] allele has a minor functional defect, while p.[Arg74Trp;Val201Met;Asp1270Asn] causes mild CF [14].

The presence of complex alleles complicates not only the diagnosis of CF, but also the choice of targeted therapy for patients with CF. Complex alleles of the *CFTR* gene can decrease the total amount of CFTR protein available to targeted drugs with the simultaneous presence of several variants on one allele (for example, due to increased proteolysis CFTR or disruptions of protein folding mechanisms) or by changing CFTR three-dimensional configuration and screening its binding site to CFTR-modulators. A similar example is the complex allele p.[Ile148Thr;Gly378Ter] ([I148T;G378X]) [12].

The frequency of complex alleles is currently underestimated. Often, after the detection of p.Phe508del or some other frequently occurring variant of *CFTR*, no additional cis variants were searched, and therefore the frequency of complex alleles is considered low. Due to the fact that additional studies on the carrier of complex alleles have not been conducted before, diagnosis may be solely based on the detection of two variants in the trans position. This approach can lead to a significant distortion of the clinical picture in cases where a complex allele remains undetected. In such instances, an undetected complex allele may be interpreted as single variant even if two or more of them are present in the cis state.

### Origins of Cis Variants—The First Cases of Studying Complex Alleles of the CFTR Gene

Complex alleles of the *CFTR* gene and their phenotypic manifestations were described for the first time in 1991 [15]. The analysis of biological samples of parents with CF showed that on the maternal chromosome, in addition to p.Phe508del, the p.Arg553Gln (R553Q) variant was present, whereas the paternal chromosome carried the p.Arg553Ter (R553X) mutation. Thus, a patient with the *CFTR* genotype p.[Phe508del;Arg553Gln]/p.Arg553Ter became the first known CF patient to inherit three sequence changes in the *CFTR* gene. The authors of the work described as phenotypic features of a patient with the genotype p.[Phe508del;Arg553Gln]/p.Arg553Ter a normal sweat test index in combination with typical manifestations of cystic fibrosis [15]. A similar case was described in the literature 18 years later, in 2009 [16]. This is a case report of a newborn from the Netherlands who suffered from growth retardation, steatorrhea, chronic cough, and external secretory pancreatic insufficiency. In early childhood, false negative results of sweat tests were recorded in the patient. As a result of the studies conducted on cell models, it was concluded that the p.Arg553Gln variant partially eliminates the pathological consequences of the F508del variant, but does not completely prevent the phenotypic manifestations of the disease [16]. In the period from 1992 to 1995, three more complex alleles containing missense mutations were described [17,18,19].

## 2. Regional Prevalence of Complex Alleles

### 2.1. Investigation of Complex Alleles p.[Leu467Phe;Phe508del] and p.[Ser466X;Arg1070Gln] Common in Russian CF Patients

In the Russian Federation, in 2022, a mass screening of the *CFTR* gene was conducted among 810 patients aged 2 to 39 years with a diagnosis of “cystic fibrosis caused by a mutation of p.Phe508del in a homozygous state” by MLPA. In 8.02% (65 out of 810 patients), a variant of the nucleotide sequence p.Leu467Phe (L467F) in the cis position with the variant p.Phe508del was present in the genotype. At the same time, in 0.49% (4 patients), the variant p.Leu467Phe was detected in a homozygous state, which indicates the presence of a complex allele, p.[Leu467Phe;Phe508del], on both chromosomes. At the same time, the clinical course of the disease in groups with genotypes p.Phe508del/p.[Leu467Phe;Phe508del] and p.Phe508del/p.Phe508del did not differ in age of diagnosis, characteristics of respiratory function, nutritional and microbiological status, frequency of exacerbations, or therapy, including courses of intravenous antibacterial therapy [20,21].

The second most common complex allele in the Russian Federation is p.[Ser466X(TGA);Arg1070Gln] ([S466X;R1070Q]) [4]. In 2020, 41 cases of this variant with severe phenotypic manifestation were recorded in the Register in the Russian Federation [4]. The study of the phenotypic manifestations of the complex allele allowed to conclude that this mutation belongs to Class 1 and has a severe phenotype [22]. Additional studies to assess the functional activity of the CFTR channel conducted on intestinal organoids confirmed the absence of an organoid swelling to forskolin and targeted drugs for [S466X;R1070Q] [23]. Thus, this complex allele can be attributed to a Class 1 mutation with a high degree of probability [22,23,24].

### 2.2. The Complex Allele p.[Ala238Val;Phe508del] Is Enriched in Italian CF Patients

In Italy, 218 patients were examined who, at the first analysis, turned out to be homozygous for the p.Phe508del mutation (n = 63) or heterozygotes with the F508del variant and any other pathogenic variant in the second allele of the *CFTR* gene (n = 155). A total of 18 (7 among homozygotes and 11 among heterozygotes) patients had the cis variant p.Ala238Val (A238V) as part of the complex allele p.[Ala238Val;Phe508del] ([A238V;F508del]) [25]. Consequently, the frequency of occurrence of the complex allele p.[Ala238Val;Phe508del] among patients with the variant p.Phe508del was equal to 8.25% (the allelic frequency is 0.04). Based on the obtained clinical data, the authors concluded that carriers of the complex allele p.[Ala238Val;Phe508del] have lower sweat chloride indicators compared to p.Phe508del/p.Phe508del patients.

However, lung function is slightly reduced in patients with a complex allele, and they have a higher inflammatory response compared to homozygotes p.Phe508del/p.Phe508del. Nevertheless, it was concluded that the complex allele of p.[Ala238Val;Phe508del] seems to be associated with fewer common complications than in the control groups of p.Phe508del homozygotes. These results are important for the correct assessment of the clinical response in patients with the complex allele p.[Ala238Val;Phe508del] when analyzing the effect of new drugs for the treatment of CF [25].

### 2.3. Complex Alleles Are Enriched in French CF Patients

A common complex allele in northwestern France (Brittany) is p.[Phe508del;Ile1027Thr] ([F508del;I1027T]) [13]. This complex allele is detected with a frequency of >5% of chromosomes carrying the mutation p.Phe508del. The p.Ile1027Thr mutation is a polymorphism that does not cause CF on its own [26].

El-Seedy et al. conducted a large-scale study of the pathogenicity of variants of the *CFTR* gene p.Gly149Arg (G149R), p.Asp443Tyr (D443Y), p.Gly576Ala (G576A), and p.Arg668Cys (R668C), both single variants and those included in different complex alleles. Out of a total of 1423 people examined in four molecular genetic laboratories in France, 153 people were identified as carrying at least one of these variants both separately and as part of a complex allele. Epidemiological data were collected for 153 patients, genotype-phenotype correlation was performed, and the structural and functional features of complex variants of p.[Gly576Ala;Arg668Cys], p.[Gly149Arg;Gly576Ala;Arg668Cys], and p.[Asp443Tyr;Gly576Ala;Arg668Cys] were studied.

CF with a severe clinical picture was observed in three patients carrying p.Gly149Arg as part of the complex allele p.[Gly149Arg;Gly576Ala;Arg668Cys] in the trans position with a severe *CFTR* variant. This genotype was not detected in healthy people or in people with a mild form of the disease, which suggests that the main aggravating variant in the complex allele is p.Gly149Arg, which can be attributed to Class 2. Functional studies have shown that p.Gly149Arg causes serious disruptions of CFTR protein processing and the lack of the functional chloride channel on the plasma membrane, p.Asp443Tyr affects protein maturation, p.Gly576Ala causes a disorder of splicing, and p.Arg668Cys has a virtually mildly effect on the functioning and structure of the CFTR chloride channel. According to the results of the study, it was concluded that the presence of a triple complex allele p.[Asp443Tyr;Gly576Ala;Arg668Cys] is accompanied by a decrease in CFTR function and is responsible for the occurrence of CFTR-related disorders (CFTR-RD: bronchiectasis, chronic sinusitis, idiopathic pancreatitis, cholestasis, and congenital bilateral absence of vas deferens (CBAVD)) [27].

### 2.4. Complex Allele [c. 744-33GATT(6);869+11C>T;p.Asn1303Lys] among Lebanese CF Patients

In 2015, the *CFTR* gene was screened among Lebanese patients carrying the pathogenic variant p.Asn1303Lys (N1303K, severe Class 2 variant), which is most commonly found in Mediterranean countries. Using direct automated sequencing, hybrid minigene constructs to study splicing, and RT-PCR, Farhat et al. examined seven Lebanese families whose genetic diagnosis included the p.Asn1303Lys variant and a pathogenic variant on the second parental allele in trans. It was found that the p.Asn1303Lys variant in all the studied patients was associated with a complex allele c.[744-33GATT(6);869+11C>T] in the cis position with the formation of a triple complex allele, [c.744-33GATT(6);869+11C>T;p.Asn1303Lys]. Using minigene constructs, the effect of a single mutation of p.Asn1303Lys on the splicing process was not revealed, while the complex allele induces minor skipping of exon 7 [28].

### 2.5. Complex Allele p.[Gly253Arg;Gly451Val] among Ecuadorian CF Patients

In Ecuador, 141 unrelated patients with CF (71 men and 70 women) aged 3 months to 32 years were examined. The complex allele p.[Gly253Arg;Gly451Val] ([G253R;G451V]) was identified in six patients, including one homozygote and five compound heterozygotes. The analysis of p.[Gly253Arg;Gly451Val] in silico predicted its severe pathogenicity with high probability. The missense variant of p.Gly253Arg not as part of a complex allele was found only in one person with the genotype p.Gly253Arg/p.Ala75Glu (p.A75E) [29]. A patient with the genotype p.Gly451Val/p.Phe508del was clinically diagnosed with CF, which indicates the pathogenicity of p.Gly451Val [30].

### 2.6. Complex Allele p.[Arg334Trp;Arg1158Ter] in a Family of Portuguese CF Patients

In 1996, the group Duarte et al. conducted a study on a family of Portuguese origin, consisting of three generations, with two brothers who were diagnosed with CF at the ages of 3 and 8 years on a positive sweat test. After DNA diagnostics, the genotype p.[Arg334Trp;Arg1158Ter]([R334W;R1158X])/p.Phe508del was established in the patients. The disease was characterized by a decrease in lung function and infection with *Pseudomonas aeruginosa*, while the function of the pancreas was preserved. At the age of 13, the older brother unexpectedly developed a severe respiratory disease, which 5 months later led to death from cardiorespiratory insufficiency. It was shown that the corresponding complex allele p.[Arg334Trp;Arg1158Ter] mRNA was present, but in a reduced amount. The authors suggest that the combination of the mild missense variant p. Arg334Trp with the nonsense variant p.Arg1158Ter (which can lead to the formation of a functional, but truncated CFTR protein) does not lead to a complete loss of CFTR function and therefore causes mild phenotypic manifestations [31].

Thus, *CFTR* complex alleles occur at different frequencies in different regions, which is related to the origin of the ethnic groups. This is also observed in cases of variants in the trans position, without a complex allele [32]. For example, among Russian patients with CF, there are two pathogenic variants in trans in the *CFTR* gene which are enriched in patients from this region: CFTRdele2,3 and E92K (6.15% and 3.25%, respectively) [4,32]. This pattern also remains true for *CFTR* complex alleles.

## 3. The Influence of Genetic Variants in the Composition of a Complex Allele on Its Pathogenicity

Each variant in the complex *CFTR* allele can affect its overall pathogenicity. Most often, the pathogenicity of *CFTR* variants increases when they are combined in a complex allele, or the variants are neutral, and in this case, in a complex allele, only one variant causes the disease, and the other does not affect the course and phenotypic signs of CF. There are rarer cases when one variant in a complex allele reduces the pathogenicity of the second and facilitates the course of the disease (Figure 1).

Currently, it is known that additional variants in the composition of complex alleles can distort the idea of the pathogenicity of an already known variant previously entered into the *CFTR* database as causing CF. But in reality, such variants may have a residual or normal function of the CFTR protein and be in a cis position with another pathogenic variant that has not been identified, and may be located in another exon or in distant cis-regulatory elements. For example, Romey et al. described the case of a complex allele [c.-234T>A;p.Ser549Arg] ([c.-234T>A;p.S549R]) variant of c.-234T>A which enhances the activity of the *CFTR* promoter, reducing the severity of the phenotype caused by p.Ser549Arg, causing CF [33]. This phenomenon illustrates the potential therapeutic possibility of enhancing *CFTR* expression to reduce the severity of the disease and its “lighter” course.

The variant p.Arg117His (R117H) is probably always associated with two intron 8 splice variants T5 and T7 [34,35,36]. According to a French study, the penetrance of CF in individuals with the [p.Arg117His;T7]/p.Phe508del genotype was 0.03%, and the manifestation of CFTR-RD (CBAVD) was 3% [36]. The T7 variant is considered neutral, but the T5 variant was shown to affect exon 9 splicing reducing the amount of CFTR protein, probably causing a more severe phenotype [35]. Thus, people with [p.Arg117His;T7] will produce normal levels of partially functional CFTR, which can cause CBAVD or (rarely) CF (97% [p.Arg117His;T7]/p.Phe508del—without symptoms of the disease). This differs from individuals, bearing [p.Arg117His;T5], which leads to a more severe disease phenotype [35].

In an article by Schucht et al. in brothers with the genotype p.Phe508del/p.[Arg74Trp;Val201Met;Asp1270Asn] ([R74W;V201M;D1270N]), the effect of a complex allele on CFTR function was studied using the method of intestinal current measurements (ICM) on rectal biopsies, from which samples were then obtained to study the amount and processing of CFTR protein by immunoblotting. The phenotypic manifestations of CF in the older child were more pronounced than in the younger one. The results of ICM (analysis of the functional activity of CFTR) and Western blot (the amount of CFTR protein) are consistent with the different clinical phenotype of the brothers. Normal (younger brother) and subnormal (older brother) chloride secretion measured by ICM were associated with normal CFTR protein formation and a fourfold decrease in its amount, respectively. Thus, the authors found differences in the same family in phenotypic manifestations with the same genotypes [37].

In addition to complex alleles with pathogenic variants, in which one of the variants “softens” the pathogenicity of the second and leads to a less severe course of CF [31,33,34], a number of complex alleles have been identified in which two or more non-pathogenic or “mild” variants in the cis position cause clinical manifestations of CF or “weigh down” the phenotype (Figure 1). The main mechanism is a decrease in the normal amount of functional CFTR protein. For example, in twins with the genotype p.Phe508del/p.[Arg347His;Asp979Ala] ([R347H;D979A]), a severe course of CF was recorded: pancreatic insufficiency, expressed pulmonary manifestations with generalized infiltration, and structural changes in the lungs [38]. Based on clinical data, Clain J. et al. decided to conduct studies of the complex allele p.[Arg347His;Asp979Ala] on HeLa cells transfected with pTCFwt plasmid [39]. Using structural and functional analysis, it was shown that the amount of mature protein in mutant cells is 59% of the norm. It has been shown that the complex allele p.[Arg347His;Asp979Ala] causes improper processing (glycosylation) due to the p. Asp979Ala variant and a decrease in the functional activity of the chloride channel caused by the p. Arg347His variant. Option p.Asp979Ala is responsible for the defective processing of p.[Arg347His;Asp979Ala]-CFTR, since it is the negative charge on the residue of 979 that is necessary for the proper processing of CFTR. The lack of a sufficient amount of functional CFTR protein leads to the appearance of a severe phenotype in patients with this complex allele [39].

The pathogenicity of the variant p.Leu997Phe (p.L997F) was also studied in two different allelic contexts: either as an independent variant, or in combination with p.Arg117Leu (R117L). The evaluation of the phenotypic manifestations of CF showed that p.Leu997Phe alone can cause a “mild” phenotype of CF or CFTR-RD, while as part of the complex allele p.[Arg117Leu;Leu997Phe], it causes a characteristic clinical picture of the disease: four of the twelve compound heterozygotes with p.[Arg117Leu;Leu997Phe] were characterized by severe and mild courses of the disease [40]. Separately, p.Arg117Leu causes CFTR-RD. Thus, in patients with the complex allele p.[Arg117Leu;Leu997Phe], there is a deterioration in the clinical manifestations of CF under the influence of the complex allele compared with patients with only p.Leu997Phe and p.Arg117Leu (in combination with the pathogenic variant on the second parent allele) [40].

A clinical case of CF in one family with three children was also described. DNA analysis showed the presence of the paternal mutation p.Asn1303Lys in the children, and the maternal *CFTR* allele turned out to be complex with three missense mutations: p.Asp443Tyr, p.Gly576Ala, and p.Arg668Cys (R668C) [41]. Each of the variants was previously recorded separately in men with CBAVD [27,42]. The manifestations of the disease were different among the three siblings: the eldest child was an asymptomatic carrier of the genotype with a complex allele, while the middle child had a history of lower respiratory tract infection and slightly elevated levels of chlorides in sweat. During the mother’s pregnancy with the third child, hyperechogenicity of the fetal intestine was recorded. The family was consulted about atypical CF with a mild clinical course, at least in childhood, as it was observed in the other brothers, with the possibility of a likely mild manifestation of lung complications associated with CF at a later age in any of the children [41].

The literature describes cases when a neutral variant of the nucleotide sequence, located in the cis position, contributed to the emergence of a severe CF phenotype. French researchers have studied the complex allele p.[Ser912Leu; Gly1244Val] ([S912L;G1244V]) associated with CF [43]. To assess the contribution of each variant to the phenotypic manifestations of the complex allele, site-directed mutagenesis was used. The *CFTR* gene was expressed in HeLa cells, after which the processing and activity of the CFTR chloride channel were analyzed. In the case of the variant p.Gly1244Val and the complex allele p.[Ser912Leu;Gly1244Val], protein processing (the level of glycosylated protein) was normal, but the chloride channel function markedly decreased compared to the wild type. At the same time, the CFTR function in the case of the complex allele p.[Ser912Leu;Gly1244Val] was significantly lower compared to p.G1244V (activity of the CFTR channel WT>p.Gly1244Val>p.[Ser912Leu;Gly1244Val]). Cultures with the nucleotide variant p.Ser912Leu were characterized by normal protein processing and an unchanged (normal) function of the CFTR channel. Thus, it was concluded that the p.Ser912Leu variant should be considered as a polymorphism in isolation, but it can significantly disrupt the CFTR function when inheriting the p.Gly1244Val variant in the cis position [43]. Such an increase in clinically insignificant variants of *CFTR* (including polymorphisms) of each other’s pathogenicity as part of a complex allele, which leads to a decrease in the activity of the chloride channel, has also been described in other studies [39,44,45].

A case report was also described when a 6-year-old girl of Albanian origin without a burdened family history was diagnosed with CF. A severe phenotype was observed: meconium ileus, chronic constipation, cough, bone age delay, hepatomegaly, pancreatic malabsorption, andlung atelectasis. Conducting NGS allowed to establish the presence of a cis variant of p.[Glu474Lys;Phe508del] ([E474K;F508del]) inherited from the mother. Further bioinformatic analysis revealed the deletion of 20 and 21 exons inherited from the father (c.3158_3468+3219del). It is not possible to definitively assess the pathogenicity of the p.Glu474Lys variant; however, presumably it is pathogenic [46].

In 2016, a study was conducted on a cohort of patients with CF, in whose genotype complex *CFTR* alleles were identified, establishing a correlation between the genotype, phenotype, and the results of in vitro and ex vivo studies [14]. Among the complex variants were the following: p.[Ile148Thr;Ile1023_Val1024del] ([I148T;3199del6]), n = 5; p.[Arg74Trp;Val201Met;Asp1270Asn], n = 8; p.[Arg74Trp;Asp1270Asn], n = 4; [1210-34TG[12];1210-12T[5];p.Glu92Lys], n = 3; where n is the number of patients. The potential-dependent (gating) activity of the CFTR channel on the nasal epithelial cells (NEC) of patients with complex alleles was analyzed in comparison with patients with homo- and heterozygotes without complex alleles, and 70 people participated in the study. The complex allele p.[Ile148Thr;Ile1023_Val1024del] caused CF with a severe clinical picture in five heterozygotes with a Class 1–2 variant in the second parent allele. The activity of CFTR in these patients’ ex vivo on NEC was comparable to the activity of the CFTR channel in patients with two Class 1–2 variants. A complex allele with two variants of p.[Arg74Trp;Asp1270Asn] had virtually no effect on the activity of the CFTR channel, while p.[Arg74Trp;Val201Met;Asp1270Asn] caused mild CF in four out of five patients also carrying a Class 1–2 variant in the trans position. Expression in heterologous systems showed that the variant p.Arg74Trp behaves as a polymorphism, a double mutant p.[Arg74Trp;Asp1270Asn] demonstrates a reduced transport of chloride ions (a reduced function of the CFTR channel), and in systems with triple mutation p.[Arg74Trp;Val-201Met;Asp1270Asn], CFTR conductivity also decreases, and post-translational modification (glycosylation) and protein transport to the membrane are disrupted [14]. Compound-heterozygous patients with the allele [1210-34TG[12];1210-12T[5];p.Glu92Lys] and the Class 1–2 variant had either a diagnosis of CF with the “mild” phenotype, or CFTR-RD [14].

Among patients with CF in the USA, the Ile148Thr is a common *CFTR* variant. Using the example of three complex alleles, namely, [p.Ile148Thr;T9], [p.Ile148Thr;T7], and p.[Ile148Thr;Ile1023_Val1024del], it was shown that p.Ile148Thr is neutral and does not affect the pathogenicity of additional variants in the composition of complex alleles. At the same time, carriers of the p.Ile148Thr variant in combination with the pathogenic *CFTR* variant on the second allele showed no signs of CF or CFTR-RD [47].

There are cases of non-pathogenic complex alleles of the *CFTR* gene. In 2004, Claustres et al. investigated the presence of cis variants in the *CFTR* gene in 437 families with an established diagnosis of CF and 170 families with isolated azoospermia caused by CBAVD syndrome. As a result, it was shown that the complex allele p.[Arg74Trp;Asp1270Asn] in combination with p.Pro67Leu (P67L) on the second parent allele does not cause disease, from which it was concluded that this complex allele is not enough to cause disease [48].

The wave of the COVID-19 epidemic also left a mark in the study of the influence of this disease on the clinical and molecular manifestations of CF. In particular, a 2022 study reports a sample consisting of 585 people with a positive PCR result for SARS-CoV-2. Using genome-wide sequencing methods and methods for evaluating CFTR channel activity, the group of Baldassarri et al. found that carriers of one pathogenic allele of the *CFTR* gene have a higher risk of severe COVID-19 and 14-day death compared to people without pathogenic variants of the *CFTR* gene. The machine learning post-Mendelian model defined *CFTR* as a bidirectional modulator of COVID-19 outcomes. A rare complex allele, p.[Gly576Val;Arg668Cys] ([G576V;R668C]), was discovered, which determines a milder course of the disease due to gain-of-function mechanism *CFTR*. The work also characterized complex *CFTR* alleles, including p.[Ala238Val;Phe508del], p.[Ala238Val;Phe508del], p.[Arg74Trp;Val201Met;Asp1270Asn], p.[Ile1027Thr;Phe508del], and p.[Ile1506Val;Asp1168Gly] ([I506V;D1168G]), and the simple variants p.Arg347Cys (R347C), p.Phe1052Val (F1052V), p.Tyr625Asn (Y625N), p.Ile328Val (I328V), p.Lys68Glu (K68E), p.Ala309Asp (A309D), p.Ala252Thr (A252T), p.Gly542Ter (G542X), p.Val562Ile (V562I), p.Arg1066His (R1066H), p.Ile506Val (I506V), and p.Ile807Met (I807M), which lead to a decrease in CFTR function and, thus, to a more severe COVID-19 course. Thus, the accurate genetic analysis of *CFTR* is an important tool for identifying patients at risk of severe COVID-19 [49].

## 4. Effectiveness of Targeted Therapy in Patients with Complex Alleles of the *CFTR* Gene

Terlizzi et al. conducted a study of the effectiveness of targeted therapy for patients with genotypes p.[Ala238Val;Phe508del]/p.Asp1152His (D1152H) and p.[Phe508del;Ile1027Thr]/p.Arg709Term ([F508del;I1027T]/R709X). The patients had external secretory pancreatic insufficiency (exocrine pancreatic insufficiency). The study reports clinical data for patients before and after 4 weeks of treatment with tezacaftor/ivacaftor, lumacaftor/ivacaftor, and elexacaftor/tezacaftor/ivacaftor. Four weeks of treatment for a patient with p.[Ala238Val;Phe508del]/p.Asp1152His, led to the normalization of the sweat test and improvement of FEV_1_ (+7 points) (from 80% to 87%), and an increase in BMI (+0.85) and CFQ-R (+22.2 points). A patient with p.[Phe508del;Ile1027Thr]/p.Arg709Term (R709X) had an improvement in the indicators of the sweat test, FEV_1_ (+9 points) (from 61% to 70%), and CFQ-R (+16.7 points) after the use of targeted therapy [50].

Most patients with CF are carriers of the p.Phe508del variant, which affects the folding, processing, function, and stability of the CFTR protein. Treatment aimed at increasing CFTR function in p.Phe508del homozygotes using lumacaftor/ivacaftor showed efficacy and a very wide variability of responses among patients [51]. Such variability is associated with the destabilization of p.Phe508del-CFTR with a prolonged use of ivacaftor [52,53] or the presence of cis variants [54]. The presence of complex alleles may be the reason for the lack of response to lumacaftor/ivacaftor in patients homozygous for p.Phe508del.

Previously, it was found that the p.Leu467Phe variant does not lead to the development of CF, but reduces the amount of functional (fully glycosylated) CFTR by two times compared to the norm [55], but p.Leu467Phe in combination with p.Phe508del causes a lack of response to treatment with CFTR correctors [20]. Variants of p.Leu467Phe and p.Phe508del have an additive pathogenic effect, which can no longer be eliminated by the lumacaftor corrector. Both mutations are localized in the nucleotide binding domain 1 (NBD1) and lead to protein folding defects that prevent protein maturation [55]. These results illustrate that complex alleles affecting the folding of the CFTR protein can cause resistance to tezacaftor therapy. The functional CFTR protein is also not restored by tezacaftor/elexacaftor [21]. It is also proved on the model of intestinal organoids that for patients with the genotype p.Phe508del/p.[Leu467Phe;Phe508del], triple therapy elexacaftor/tezacaftor/ivacaftor will be effective in clinical practice [21]. However, in a study by Terlizzi et al. for a patient with the same genotype (p.Phe508del/p.[Leu467Phe;Phe508del]) before and after 4 weeks of therapy with tezacaftor/ivacaftor, lumacaftor/ivacaftor, and elexacaftor/tezacaftor/ivacaftor, positive changes were observed in the clinical manifestations of CF [50]. The authors evaluated the effectiveness of therapy using a 6 min walk test and an assessment of lung function before and after therapy, and the conclusion about the ineffectiveness of therapy was drawn based on the lack of improvement in this parameter. However, the authors also note that the effect was not observed only by this criterion [50]. The search for a variant of p.Leu467Phe is crucial in Russian patients with cystic fibrosis with p.Phe508del who do not respond to therapy with thezacaftor/ivacaftor and lumacaftor/ivacaftor, given the high frequency of the complex allele of p.[Leu467Phe;Phe508del]) [20].

The presence of an unidentified complex allele can reduce the amount of full-sized glycosylated CFTR, which is a target for the corrector, or affect protein folding, and, as a consequence, the effectiveness of the corrector by reducing the availability of the binding site (Figure 1). In 2018, an NGS screening of a cohort of patients consisting of 36 people was carried out. Among the studied patients, a genotype p.[Phe87Leu;Phe508del;Ile1027Thr]/p.Phe508del with a complex allele was found in four people. Before and 6 months after the start of ivacaftor/lumacaftor treatment, the following biomarkers of CFTR activity were measured: sweat test bioassays, NPD, and ICM. The results showed that the patients did not improve their clinical condition 6 months after the start of treatment. In the presence of a complex allele, p.[Phe87Leu;Phe508del;Ile1027Thr] ([F87L;F508del;I1027T]), there is no response to ivacaftor/lumacaftor in a cohort of patients with the second variant p.Phe508del in trans; probably additional exon variants p.Phe87Leu and p.Ile1027Thr can affect protein correction, which reduces the function of the CFTR channel and ivacaftor/lumacaftor therapy is ineffective [54].

The mechanisms for reducing the effectiveness of targeted drugs in the presence of complex alleles are as follows:-Reduction of the total amount of functional fully glycosylated CFTR protein.

Cases with a disruption of the folding of the CFTR protein are described for the following complex alleles, including a variant of Phe508del, in the literature and databases of the CF (cftr1.org, cftr2.org): p.[Leu53Val;Phe508del] ([L53V;F508del]), p.[Phe87Leu;Phe508del;Ile1027Thr], p.[Leu130SerfsX4;Phe508del] ([Leu130;F508del]), and p.[Ile203;Phe508del]. Conversely, variant c.-234T>A enhances the activity of the *CFTR* promoter, which mitigates the phenotypic manifestations of variant p.Ser549Arg [34] and increases the amount of protein available for interaction with CFTR modulators. If there are options c.609C>T (p.Ile203) and c.2770G>A (p.Asp924Asn; D924N), there is an increase in exon skipping during splicing. Thus, the two described variants, being in the cis position with p.Phe508del, reduce the amount of CFTR protein available for targeted therapy [13,55].

Sondo E. et al. used ex vivo nasal epithelium models to study the effect of elexacaftor/tezacaftor/ivacaftor on the function of the CFTR channel in two patients carrying the complex allele p.[Leu467Phe;Phe508del] in a compound heterozygous position with Class 1 mutations G542X and E585X. The results showed no response when exposed to a triple combination of CFTR modulators (elexacaftor/tezacaftor/ivacaftor). Using heterologous protein expression and Western blot, the authors found that the complex allele p.[Leu467Phe;Phe508del] leads to degradation of misfolded CFTR by endoplasmic-reticulum-associated protein degradation (ERAD). The use of CFTR modulators elexacaftor/tezacaftor/ivacaftor does not increase the amount of the mature fully glycosylated form of CFTR, and therefore does not affect the functional activity of p.[Leu467Phe;Phe508del]-CFTR protein [56].

-Preventing the interaction of the modulator with the binding site on the CFTR protein by changing its three-dimensional configuration and screening the binding site (folding disruption).

Another mechanism by which cis variants affect the effectiveness of CFTR modulator therapy is not associated with a decrease in the amount of CFTR protein, but with a change in the protein binding site with the CFTR modulator. For example, the VX-809 (lumacaftor) and VX-661 (tezacaftor) corrector binding site is a multidomain pocket near residues F374-L375; all modifications at this region can affect the CFTR correctors’ interaction [53]. Interestingly, that F87 amino acid is located near the putative VX-809 binding site and may influence lumacaftor binding, and the p.Phe87Leu variant prevented CFTR function rescue by lumacaftor. Therefore, pathogenic cis variants may identify important regions involved in CFTR sensitivity to modulators [13,53].

Thus, additional cis variants may affect the development of CF and its severity, as well as the sensitivity of the CFTR channel to modulators (Table 1).

### Solving the Problem of Selecting Targeted Therapy in the Presence of Complex Alleles of the CFTR Gene

The evaluation of the effectiveness of CFTR modulators is based on determining their ability to increase the amount of functional CFTR protein on the surface of epithelial cells. In a number of European countries, 3D cultures of intestinal organoids obtained from patients’ own intestinal tissue have become in vitro models for studying the disruption of the chloride channel in any genetic variant of the *CFTR* gene, the personalized assessment of the residual activity of the CFTR protein, and the choice of targeted therapy for CF [59,60]. Intestinal organoids are multicellular three-dimensional (3D) closed structures consisting of a single-layer epithelium and are simplified versions of the intestine obtained in vitro. In 2011, human intestinal organoids were obtained for the first time [61]. Compared to standard models (immortalized cell lines and animals), organoids (1) open up the possibility for personalized medicine; (2) simulate the structure of the intestinal epithelium, which is formed by several types of cells; and (3) help solve the problem of species-specific differences. Moreover, (4) the cultures of intestinal organoids are permanent (replantable) and can be maintained for years [62,63,64].

The forskolin-induced swelling (FIS) assay on intestinal organoids is a simple and convenient method for determining the degree of dysfunction of the CFTR channel and the effectiveness of targeted drugs in patients with any genetic variants of *CFTR*, including complex alleles. The principle of the FIS assay is based on the direct activation by forskolin of the enzyme adenylate cyclase in intestinal epithelial cells, an increase in the concentration of intracellular cAMP, and an increase in the conductivity of CFTR. With a normal ionic conductivity of CFTR (organoids from healthy people), its activation leads to fluid secretion into the lumen of the organoids and to their rapid swelling. If the CFTR protein is not formed or is not functional (depending on the variant), the organoids do not swell. After treatment with forskolin, visual assessment (microscopy) and a quantitative analysis of changes in the size of organoids are performed [60,65]. The FIS of intestinal organoids depends only on the conductivity of the CFTR protein and directly reflects its residual function.

The first examples of the study of the effect of CFTR modulators on the restoration of the function of the chloride channel in patients with CF in the presence of complex alleles in the genotype are given in the works of Russian researchers on the model of intestinal organoids [20,21,22,23,24,66]. The effectiveness of this approach has been proven in real clinical practice.

## 5. Summary

Complex alleles contain at least two variants of the *CFTR* gene sequence in the cis position, and each pathogenic variant can affect individual stages of CFTR biogenesis. Variants in the complex *CFTR* allele are often polymorphisms that do not lead to the occurrence of CF, but their combined negative effect is pathogenic, provoking clinical manifestations of the disease. There are also opposite effects, when the course of CF becomes milder in the presence of several cis variants, including pathogenic.

Preliminary data show that the frequency and regional prevalence of complex alleles may vary in different populations. In this regard, the sequencing method is becoming increasingly relevant when conducting DNA diagnostics to determine the carrier of complex alleles. The inclusion of the inference gene sequencing in the neonatal screening algorithm may help solve this problem. It is necessary to carefully assess the clinical condition of patients with CF: if patients with the same established *CFTR* genotypes have serious differences in the manifestation of the disease, then additional extended molecular studies should be carried out to exclude complex alleles in those patients.

Complex alleles can negatively affect the effectiveness of treatment with CFTR modulators, which must be taken into account when selecting the optimal therapy. In case of detection of a complex allele in patients with CF, it is recommended to conduct a forskolin-induced swelling assay on intestinal organoids for a personalized assessment of the effectiveness of targeted drugs.

To summarize, we can draw the following main conclusions:Further studies are needed to determine the frequency of complex alleles in different populations to determine their effect on the course of CF and the effectiveness of targeted therapy.The use of sequencing can contribute to the early detection of carriers of complex alleles.The presence of several variants in the cis position most often leads to a decrease in the CFTR function. Sometimes, the introduction of additional variants (especially in the promoter regions of the gene) can be considered as a potential way to enhance the function of CFTR.In the absence of a response to targeted therapy (if there is a genetic variant in the instructions for targeted drugs), it is necessary to sequence the *CFTR* gene for the presence of complex alleles and conduct a FIS assay on intestinal organoids.

## Figures and Tables

**Figure 1 ijms-25-00114-f001:**
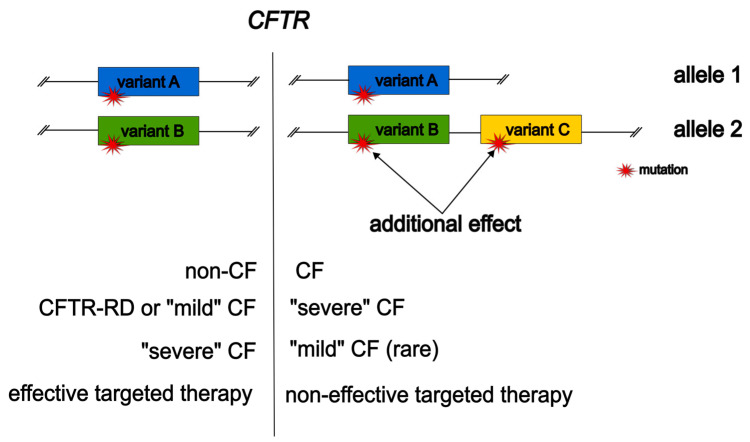
Additional cis variants lead to changes in the phenotypic manifestations of CF and the effectiveness of targeted therapy.

**Table 1 ijms-25-00114-t001:** Effect of complex alleles on the effectiveness of targeted therapy.

#	Complex Allele	Pathogenicity of Variants	Impact on the Development and Severity of CF	Genotype	Efficacy of Targeted Therapy	Epidemiology	Links
1	[R117H;TG12T5]	Both cause CFTR-RD	Cause CFTR-RD	Non-CF	Not enough information	Caucasian	[13,34]
2	[R117L; L997F]	R117L may cause CFTR-RD	Yes Increase each other’s pathogenicity	Mild	Not enough information	Not enough information	[40]
3	[R334W;R1158X]	Both cause CF	Not enough information	Severe genotype in trans with F508del	Not enough information	Portuguese	[31]
4	[I148T;G378X]	I148T does not cause CF	Not enough information	Severe genotype in trans with L867X	Not enough information	Caucasian	[12]
5	[I148T;9T;3199del6]	I148T does not cause CF, the other two cause CF	Insufficient clinical data and follow-up period	Severe genotype in trans with Class 1–3 mutations	Not enough information	American	[47]
6	[L467F;F508del]	L467F polymorphism	Does not affect	Severe genotype in trans with F508del	Lum+Iva for L467F;F508del/F508del–no; ETI for L467F;F508del/F508del–yes	Russian	[21]
7	[L467F;F508del]	L467F polymorphism	Does not affect	Severe genotype in trans with F508del	Lum+Iva for L467F;F508del/F508del–no	French	[55]
8	[L467F;F508del]	L467F polymorphism	Not enough information	Severe genotype in trans with Class 1 mutations	ETI for L467F;F508del/class 1–no	Italian	[57]
9	[F87L;F508del;I1027T]	F87L, I1027T–polymorphism	Yes Increase each other’s pathogenicity	Severe genotype in trans with F508del	Lum for F87L;F508del;I1027T/F508del–no	French	[54]
10	[R347H;D979A]	Both cause CF	Yes Increase each other’s pathogenicity	Severe genotype in trans with F508del	Not enough information	Not enough information	[38,39]
11	[E474K;F508del]	E474K-varying clinical consequence	Yes Increase each other’s pathogenicity	Severe genotype in trans with c.3158_3468+3219del	Not enough information	Albanian	[46]
12	[F508del;R553Q]	R553Q-Variant of unknown clinical significance	Does not affect	Depending on the version in trans position	Not enough information	German	[58]
13	[G576A;R668C]	Both CFTR-RD-causing	Yes Increase each other’s pathogenicity	Mild	Not enough information	French	[27]
14	[D443Y;G576A;R668C]	All CFTR-RD-causing	Yes Increase each other’s pathogenicity	Mild	Not enough information	French	[27]
15	[G149R; G576A;R668C]	All CFTR-RD-causing	Yes Increase each other’s pathogenicity	Severe genotype in trans with Class 1–3 mutations	Not enough information	French	[27]
16	[S466X(TGA);R1070Q]	Both cause CF	Does not affect	Severe genotype in trans with Class 1–3 mutations	Lum+Iva and ETI for S466X(TGA);R1070Q/CFTRdele2,3-no	Russian	[23,24]
17	[Arg74Trp;Val201Met;Asp1270Asn]	All CFTR-RD-causing	Yes Increase each other’s pathogenicity	Mild	Not enough information	German-Moroccan	[37]

## Data Availability

Data is contained within the article.

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
