# Peer review of "The Effect of Complex Alleles of the CFTR Gene on the Clinical Manifestations of Cystic Fibrosis and the Effectiveness of Targeted Therapy"

_ijms, 2023, doi:10.3390/ijms25010114_

Round 1

Reviewer 1 Report

Comments and Suggestions for Authors

The review manuscript “The effect of complex alleles of the CFTR gene on the clinical manifestations of cystic fibrosis and the effectiveness of targeted therapy” discusses the effect of complex alleles on disease severity and effects of modulator therapy on cystic fibrosis. The manuscript is a careful summary of the literature, with interesting case reports and pertinent references. It was particularly interesting how the authors go through the presence of different complex alleles in different countries. However, there are a considerable number of concerns with the text:

Major concerns:

1. Please reconsider the division into sections. It is confusing to have sections called introduction, background, results. Introduction is the same as background. A review article cannot have results, if it is not a systematic review and there are no indications that this is a systematic review. Please remove either the conclusions or the summary. You cannot have both.

2. There is a separate section about effect of modulator therapy on complex alleles. Please collect everything on that subject in this section. It is confusing that you talk about the same subject elsewhere, for example line 149.

3. The section about different complex alleles from different countries was interesting, but please conclude or discuss what this might mean. Why are the complex alleles different?

4. Figures are completely missing. Please add schematic drawings of where the different alleles are situated and what the consequences are.

Minor concerns:

1. The language needs editing. The text is not easy to read due to excessive use of words as well as use of the wrong word. A: Line 60: “cause a violation of the conductivity of the CFTR channel”. Please choose another word than “violation”. It means that someone is hurt or abused. B: Line 495: The word “violation” is wrong here too. Choose a better word.

Please edit.

2. Please write acronyms correctly: A: Line 32: “CFTR gene (cystic fibrosis transmembrane conductance regulator)” should be changed to cystic fibrosis transmembrane conductance regulator (CFTR). B: Line 338: “NMD (nonsense-mediated mRNA decay).” Should be rewritten to nonsense-mediated mRNA decay (NMD). C: Line 350: What is CBAVD an acronym of? Please do not use acronyms without explanation.

3. Line 120: What do you mean by the sentence “The clinical picture in the case when complex alleles has not been diagnosed in its presence, will be perceived as a manifestation of not complex alleles, but one mutation, which is not a reliable fact.” Please rephrase to make the sentence understandable.

Comments on the Quality of English Language

The language is understandable, but the text is difficult to follow because of language problems. Please use editing services.

Author Response

Response to Reviewer 1 Comments

Major corrections

  1. Please reconsider the division into sections. It is confusing to have sections called introduction, background, results. Introduction is the same as background. A review article cannot have results, if it is not a systematic review and there are no indications that this is a systematic review. Please remove either the conclusions or the summary. You cannot have both.

Response 1: We appreciate your comments very much and have made adjustments to the section names based on your suggestions.

  1. There is a separate section about effect of modulator therapy on complex alleles. Please collect everything on that subject in this section. It is confusing that you talk about the same subject elsewhere, for example line 149.

Response 2: We have moved the following paragraphs to a section “Effectiveness of targeted therapy in patients with complex alleles of the CFTR gene” showing the contribution of complex alleles of CFTR gene on targeted therapy:

Lines 146-169: “Previously, it was found that the p.Leu467Phe variant does not lead to the development of CF, but reduces the amount of functional (fully glycosylated) CFTR by 2 times compared to the norm [54], but p.Leu467Phe in combination with p.Phe508del causes a lack of response to treatment with CFTR correctors [21]. Variants of p.Leu467Phe and p.Phe508del have an additive pathogenic effect, which can no longer be eliminated by the lumacaftor corrector. Both mutations are localized in the nucleotide binding domain 1 (NBD1) and lead to protein folding defects that prevent protein maturation [54]. These results illustrate that complex alleles affecting the folding of the CFTR protein can cause resistance to tezacaftor therapy. The functional CFTR protein is also not restored by a more effective combination of the correctors thezacaftor/elexacaftor [22]. It is also proved on the model of intestinal organoids that for patients with the genotype p.Phe508del/p.[Leu467Phe;Phe508del] triple therapy elexacaftor/tezacaftor/ivacaftor will be effective in clinical practice [22]. However, in a study by Terlizzi et. al. for a patient with the same genotype (p.Phe508del/p.[Leu467Phe;Phe508del]) before and after 4 weeks of therapy with tezacaftor/ivacaftor, lumacaftor/ivacaftor and elexacaftor/tezacaftor/ivacaftor, therapy with tezacaftor/ivacaftor + ivacaftor, lumacaftor/ivacaftor + ivacaftor and elexacaftor/tezacaftor/ivacaftor, no positive changes were observed in the clinical manifestations of CF [55]. The authors evaluated the effectiveness of therapy using a 6-minute walk test and an assessment of lung function before and after therapy, and the conclusion about the ineffectiveness of therapy was made based on the lack of improvement in this parameter. However, the authors also make a note that the effect was not observed only by this criterion [55]. The search for a variant of p.Leu467Phe is crucial in patients with cystic fibrosis with p.Phe508del who do not respond to therapy with thezacaftor/ivacaftor and lumacaftor/ivacaftor for the Russian population, given the high frequency of the complex allele of p.[Leu467Phe;Phe508del]) [21].” we have moved to the 492-516 lines

Lines 170-180: Sondo E. et al used ex vivo nasal epithelium models to study the effect of elexacaftor/tezacaftor/ivacaftor on the function of the CFTR channel in two patients carrying the complex allele p.[Leu467Phe;Phe508del] in a compound heterozygous position with class I mutations G542X and E585X. The results showed no response when exposed to a triple combination of CFTR modulators (elexacaftor/tezacaftor/ivacaftor). Using heterologous protein expression and Western blot, the authors found that the complex allele p.[Leu467Phe;Phe508del] leads to degradation of misfolded CFTR by ERAD (Endoplasmic-reticulum-associated protein degradation), leads to a decrease in the building of the total amount of CFTR protein, while only an immature form of the CFTR protein is created, devoid of functional activity. The use of CFTR modulators elexacaftor/tezacaftor/ivacaftor does not increase the amount of mature fully glycosylated form of CFTR, therefore, does not affect the activity of p.[Leu467Phe;Phe508del]-CFTR protein [57].” we have moved to the 546-558 lines.

  1. The section about different complex alleles from different countries was interesting, but please conclude or discuss what this might mean. Why are the complex alleles different?

Response 3: Thank you for your question! We have added the section to discuss the differences between complex alleles at lines 239-244: Thus, CFTR complex alleles occur at different frequencies in different regions, which is related to the origin of the ethnic groups in which they occur and the ethnic groups. This is also observing in cases of variants in trans position, without a complex allele. [32,33]. For example, among Russian patients with CF, there are two pathogenic variants in trans in the CFTR gene are enriched of this region: CFTRdele2,3 and E92K (6.5% and 3.25%, respectively) [4, 34]. This pattern also remains true for CFTR complex alleles.

  1. Figures are completely missing. Please add schematic drawings of where the different alleles are situated and what the consequences are.

Response 4: We have added schematic drawing.

Minor corrections

  1. The language needs editing. The text is not easy to read due to excessive use of words as well as use of the wrong word. A: Line 60: “cause a violation of the conductivity of the CFTR channel”. Please choose another word than “violation”. It means that someone is hurt or abused. B: Line 495: The word “violation” is wrong here too. Choose a better word.

Please edit.

Response 1: Dear reviewer, we concur that the term "violation" is completely incorrect to use in this case, which is why we have replaced it throughout the manuscript with 'disruption' (in the case of protein) and disorder (in the case of mRNA). We understand the importance of using accurate and appropriate terminology in scientific writing, and we are grateful for your guidance in this regard.

  1. Please write acronyms correctly: A: Line 32: “CFTR gene (cystic fibrosis transmembrane conductance regulator)” should be changed to cystic fibrosis transmembrane conductance regulator (CFTR). B: Line 338: “NMD (nonsense-mediated mRNA decay).” Should be rewritten to nonsense-mediated mRNA decay (NMD). C: Line 350: What is CBAVD an acronym of? Please do not use acronyms without explanation.

Response 2: We appreciate your valuable feedback and would like to express our gratitude for your comments. We acknowledge that our use of acronyms may have been unclear and have taken your suggestions into consideration. As per your recommendation, we have made the necessary revisions to the acronyms in question. Additionally, we wish to inform you that the abbreviation "CBAVD" (congenital bilateral absence of vas deferens) and its corresponding definition were previously mentioned in our manuscript on lines 197-198. Once again, thank you for your contribution to our work, and we hope that these revisions have improved the clarity and precision of our writing.

  1. Line 120: What do you mean by the sentence “The clinical picture in the case when complex alleles has not been diagnosed in its presence, will be perceived as a manifestation of not complex alleles, but one mutation, which is not a reliable fact.” Please rephrase to make the sentence understandable.

Response 3: We have rephrased this sentence to “The clinical picture in the case when a complex allele has not been diagnosed in the presence will be perceived as clinical manifestations of a single mutation, which is not correct” (lines 103-105) to enhance its clarity and ensure better comprehension."

Reviewer 2 Report

Comments and Suggestions for Authors

The authors summarize the knowledge from the literature regarding complex alleles of CFTR. They describe the diverse landscape of complex alleles in patient populations with CF and point out geographical differences. One of the significant aspects of this review article is that it draws attention to the relatively high frequency of these types of mutations. Furthermore, it recognizes the diagnostic challenges of cis variants and highlights the fact that conventionally used mutation-specific PCR assays underestimate the frequency of alleles carrying complex mutations. Additionally, it stresses the importance of sequencing the CFTR gene to achieve a complete evaluation of the genetic status of patients with CF. Moreover, the authors analyze the diverse modifying effect of cis variants compared to single mutations in one allele. This work provides substantial value in clinical evaluation, therapy, and research of Cystic Fibrosis.

Critics:

1)     In line 58, please add a few words regarding the molecular pathology of Class 3 variants (like include in the sentence the following (“…..lead to the formation of either critically small amounts of CFTR protein, or lead to a complete loss of protein synthesis, or impede of channel gating namely these……)). At the beginning of the sentence, class 3 mutations are mentioned, but the molecular mechanism behind the defect is not explained like it was done for class 1 or class 2.

2)     In line 73, please add the three drugs' names that are included in combination therapy (tezacaftor/elexacftor/ivacaftor) at the end of the sentence to clarify which drugs are combined from the four listed previously. This knowledge is well-known in the CF field, but readers outside of the field may not know.

3)     In line 82, please correct ”3 exons” to exon 3. I believe this variant results in exon 3 skipping and not skipping three exons.

4)     Paragraph between lines 88 and 95: Please rephrase this section. The flow of cause and effect is broken. Why are the mutations changing the CFTR protein binding site toward the modulator?

5)     The sentence in between lines 102 and 104: Please rephrase this sentence. It is hard to capture the meaning of it.

6)     In line 150: Please correct the typo in the tezacaftor name.

7)     The section between lines 153 and 157: Please revise this sentence. The addition of +ivacaftor after double therapy seems erroneous.

8)     The sentence between lines 169 and 173: Please rephrase this sentence. Building the total amount of CFTR is not a scientific term. I would instead use maturation of CFTR or degradation of misfolded CFTR by ERAD.

9)     The sentence between lines 180 and 183: The sentence sounds unfinished. Please correct it. Also, I would use organoid swelling instead of organoid response.

10)  Line 186:   I would use enriched instead of characteristic. We do not know whether this cis variant would be enriched in other groups of patients with diverse nationalities.

11)  Line 206: Same as point 10.

12)  Line 213: Please correct R668V to R668C.

13)  The paragraph between lines 250 and 258: The references seem not in alignment with the text. Please check the references. The following reference should be used instead of Ref. 29 or Ref. 32Ruiz-Cabezas JC, Barros F, Sobrino B, García G, Burgos R, Farhat C, Castro A, Muñoz L, Zambrano AK, Martínez M, Montalván M, Paz-Y-Miño C. Mutational analysis of CFTR in the Ecuadorian population using next-generation sequencing. Gene. 2019 May 15;696:28-32. doi: 10.1016/j.gene.2019.02.015. Epub 2019 Feb 11. PMID: 30763667.

14)  The sentence between lines 336 and 338: The referenced manuscript does not discuss NMD behind the p.[Arg347His;Asp979Ala] complex allele. Please correct it.

15)  Line  394: Please capitalize the code of amino acids.

16)  The sentence between lines 450 and 453: Please revise this sentence. The addition of +ivacaftor after double therapy seems erroneous.

Author Response

Response to Reviewer 2 Comments

Major corrections

  • In line 58, please add a few words regarding the molecular pathology of Class 3 variants (like include in the sentence the following (“…..lead to the formation of either critically small amounts of CFTR protein, or lead to a complete loss of protein synthesis, or impede of channel gating namely these……)). At the beginning of the sentence, class 3 mutations are mentioned, but the molecular mechanism behind the defect is not explained like it was done for class 1 or class 2.

Response 1: We have added clarifications and described in more detail the mechanism of class 3 mutation. Thank you!

  • In line 73, please add the three drugs' names that are included in combination therapy (tezacaftor/elexacftor/ivacaftor) at the end of the sentence to clarify which drugs are combined from the four listed previously. This knowledge is well-known in the CF field, but readers outside of the field may not know.

Response 2: We have added drugs combination, thank you!

  • In line 82, please correct ”3 exons” to exon 3. I believe this variant results in exon 3 skipping and not skipping three exons.

Response 3: Right, it’s exon 3 skipping, not skipping three exons. Thank you!

  • Paragraph between lines 88 and 95: Please rephrase this section. The flow of cause and effect is broken. Why are the mutations changing the CFTR protein binding site toward the modulator?

Response 4: We have rephrased this section. Thank you.

  • The sentence in between lines 102 and 104: Please rephrase this sentence. It is hard to capture the meaning of it.

Response 5: We have rephrased sentence.

  • In line 150: Please correct the typo in the tezacaftor name.

Response 6: We have corrected.

  • The section between lines 153 and 157: Please revise this sentence. The addition of +ivacaftor after double therapy seems erroneous.

Response 7: We have revised the sentence.

  • The sentence between lines 169 and 173: Please rephrase this sentence. Building the total amount of CFTR is not a scientific term. I would instead use maturation of CFTR or degradation of misfolded CFTR by ERAD.

Response 8: We appreciate your comments very much and have made correction based on your suggestions.

  • The sentence between lines 180 and 183: The sentence sounds unfinished. Please correct it. Also, I would use organoid swelling instead of organoid response.

Respond 9: We have added corrections. Thanks!

  • Line 186:   I would use enriched instead of characteristic. We do not know whether this cis variant would be enriched in other groups of patients with diverse nationalities.

Respond 10: We agree with your comment, so we used the term “enriched”. Thanks!

  • Line 206: Same as point 10.

Respond 11: We have corrected. Thanks!

  • Line 213: Please correct R668V to R668C.
    Respond 12: We have corrected. Thanks!
  • The paragraph between lines 250 and 258: The references seem not in alignment with the text. Please check the references. The following reference should be used instead of Ref. 29 or Ref. 32Ruiz-Cabezas JC, Barros F, Sobrino B, García G, Burgos R, Farhat C, Castro A, Muñoz L, Zambrano AK, Martínez M, Montalván M, Paz-Y-Miño C. Mutational analysis of CFTR in the Ecuadorian population using next-generation sequencing. Gene. 2019 May 15;696:28-32. doi: 10.1016/j.gene.2019.02.015. Epub 2019 Feb 11. PMID: 30763667.

Respond 13: We have corrected. Thanks!

  • The sentence between lines 336 and 338: The referenced manuscript does not discuss NMD behind the p.[Arg347His;Asp979Ala] complex allele. Please correct it.

Respond 14: We have corrected. Thanks!

  • Line  394: Please capitalize the code of amino acids.

Respond 15: We have corrected. Thanks!

  • The sentence between lines 450 and 453: Please revise this sentence. The addition of +ivacaftor after double therapy seems erroneous.

Respond 16: We have corrected. Thanks!

Round 2

Reviewer 1 Report

Comments and Suggestions for Authors

The review “The effect of complex alleles of the CFTR gene on the clinical manifestations of cystic fibrosis and the effectiveness of targeted therapy” gives a very nice summary of the impact of complex alleles on cystic fibrosis clinical manifestations. The authors have very nicely discussed the presence of different complex alleles in different locations. The added figure 1 also adds to clarity. Furthermore, the introduction is now divided into clearer sections. There are only some minor concerns left:

Line 104: In the presence of what? Please explain.

Line 170: “Complex alleles is enriched”. Please change to “Complex alleles are enriched”

Line 498: Please correct according to praxis: “ERAD (Endoplasmic-reticulum-associated protein degradation)” should be “Endoplasmic-reticulum-associated protein degradation (ERAD)”.

Comments on the Quality of English Language

Please proofread the language. There are some minor mistakes.

Author Response

Response to Reviewer 1 Comments

Major corrections

  1. Line 104: In the presence of what? Please explain.

Response 1: Dear reviewer! We have explained, what we meant, you can find it on lines 104-106.

(In such instances, an undetected complex allele may be interpreted as single variant even if there are two or more of them are present in the cis state)

  1. Line 170: “Complex alleles is enriched”. Please change to “Complex alleles are enriched”

Response 2: We have changed the form of the verb (Line 172). Thank you!

  1. Line 498: Please correct according to praxis: “ERAD (Endoplasmic-reticulum-associated protein degradation)” should be “Endoplasmic-reticulum-associated protein degradation (ERAD)”.

Response 3: We appreciate your comments very much and have corrected acronym according to praxis.

We also improved English of this manuscript, thank you for your comments and suggestions!
